# Synthetic evolution of herbicide resistance using a T7 RNAP–based random DNA base editor

Haroon Butt , Jose Luis Moreno Ramirez , Magdy Mahfouz

**Synthetic directed evolution via localized sequence diversification and the simultaneous application of selection pressure is a promising method for producing new, beneficial alleles that affect traits of interest in diverse species; however, this technique has rarely been applied in plants. Here, we designed, built, and tested a chimeric fusion of T7 RNA Polymerase (RNAP) and deaminase to enable the localized sequence diversification of a target sequence of interest. We tested our T7 RNAP–DNA base editor in *Nicotiana benthamiana* transient assays to target a transgene expressing *GFP* under the control of the T7 promoter and observed C-to-T conversions. We then targeted the T7 promoter-driven *acetolactate synthase* sequence that had been stably integrated in the rice genome and generated C-to-T and G-to-A transitions. We used herbicide treatment as selection pressure for the evolution of the *acetolactate synthase* sequence, resulting in the enrichment of herbicide-responsive residues. We then validated these herbicide-responsive regions in the transgenic rice plants. Thus, our system could be used for the continuous synthetic evolution of gene functions to produce variants with improved herbicide resistance.**

## Introduction

In crop plants, natural evolution and artificial selection (i.e., during domestication) relied on naturally occurring genetic variation. Increasing genetic variation in crop plants provides a basis for breeding and developing new traits of value. Genetic variation generated through random mutagenesis provides a basis for creating protein variants and useful alleles for breeding, and for evolution to develop new traits (Prohens, 2011; Hickey et al, 2019). Random mutagenesis has been used to improve crop traits; for example, radiation mutagenesis led to the generation of new traits that made the Green Revolution possible (Khush, 2001). However, radiation and chemical mutagenesis methods are not gene-specific and produce many mutations that are deleterious to the host plant.

Synthetic directed evolution relies on generating targeted or random genetic variability, followed by screening under selection pressure to identify beneficial mutations within a gene or pathway of interest (Simon et al, 2019; Morrison et al, 2020; Castle et al, 2021; Rao et al, 2021). Conventional tools for synthetic directed evolution mainly rely on in vitro diversification of a target gene, followed by screening for variants with improved functions (Smith, 1985; Chen & Arnold, 1991). These methods have been improved to generate different modifications. In single-site recombineering, exogenous oligonucleotides containing the desired sequence with flanking regions of homology are introduced to edit the target gene (Sharan et al, 2009). In the multiplexed automated genome engineering system, a library of oligonucleotides is introduced into an organism to target multiple regions of the genome, producing organisms with combinatorial diversity at the targeted regions (Wang et al, 2009). Similarly, directed evolution with random genomic mutations can be used to evolve multiple loci in a single targeted gene and its promoter regions by iteratively introducing oligonucleotides into an organism by electroporation (Nyerges et al, 2018). Although these approaches have broad applicability, they have several disadvantages, including low mutation rates and the need for labor-intensive, discontinuous steps.

These drawbacks were recently overcome by the development of in vivo mutagenesis systems that simultaneously perform gene diversification and selection to enable continuous directed evolution. The random mutation of plasmid-based genes is achieved by continuous diversification in host organisms with elevated global mutation rates, such as *Escherichia coli* strain XL1-red (Greener et al, 1997). A similar concept is used to engineer yeast via the OrthoRep system. Using this system, targeted yeast genes can be continuously diversified separately from the rest of the host genome sequence by placing these genes on orthogonal, non-genomic plasmids that replicate using error-prone DNA polymerases (Ravikumar et al, 2014, 2018).

The pioneering method of phage-assisted continuous evolution is used for the continuous evolution of proteins. Using this system, the mutational throughput in *E. coli* can be increased in vivo via a continuous process using phages, enabling the rapid evolution of gene-encoded molecules that can be linked to protein production (Esvelt et al, 2011; Simon et al, 2019; Morrison et al, 2020). However,

Laboratory for Genome Engineering and Synthetic Biology, King Abdullah University of Science and Technology (KAUST), Thuwal, Saudi Arabia

Correspondence: Magdy.mahfouz@kaust.edu.sa

this approach is best suited for phenotypes that can be linked to phage growth and cannot be directly applied to eukaryotes. These tools for directed evolution require extensive effort, long time scales, and ample resources to select for sought-after traits and are limited by the frequency and nature of mutagenesis.

To date, synthetic directed evolution has mainly been performed in single-celled organisms. Developing systems that induce localized sequence diversification at high efficiency will expand our ability to evolve traits of interest in multicellular eukaryotes (Simon et al, 2019; Hendel & Shoulders, 2021). Eukaryotic genes that have evolved in prokaryotic systems do not seem to exhibit the functions observed in the native cellular context because of non-physiological environmental conditions, the misfolding and aggregation of proteins, unintended intermolecular interactions, or unexpected modifications (Hendel & Shoulders, 2021). Therefore, establishing efficient systems for synthetic evolution within a native cellular context is essential for evolving gene functions that affect traits of interest in multicellular eukaryotes such as crop plants.

DNA base editors, including cytidine base editors (CBEs) and adenine base editors (ABEs), are powerful tools for introducing point mutations for genome engineering and synthetic evolution. Various naturally occurring cytosine deaminases, such as activation-induced cytidine deaminase (AID), rAPOBEC1, and pmCDA1 convert cytosine to uracil (Eid et al, 2018; Porto et al, 2020; Sakata et al, 2020). The U·G mismatch can be misread, resulting in C·G to T·A inversions. The base excision repair enzyme uracil DNA glycosylase reduces deamination activity by catalyzing the removal of uracil to initiate the base excision repair pathway (Krokan et al, 2002; Hegde et al, 2008). The uracil DNA glycosylase activity of CBEs is inactivated by fusing the CBE with a uracil N-glycosylase inhibitor (UGI) (Cortázar et al, 2007; Rees & Liu, 2018). ABEs do not exist in nature; instead, they were developed using a directed evolution approach. ABEs can introduce A·T to G·C point mutations into living cells (Gaudelli et al, 2017). The recently evolved ABE8e has much faster catalytic activity and offers improved editing efficiency compared with the earlier ABE10.7 variant (Lapinaite et al, 2020; Richter et al, 2020).

Targeting the base editors to the desired sequence requires programmable systems that specifically recognize the sequence of interest. CRISPR/Cas has recently been applied to diversify gene sequences of interest. For example, CRISPR-X technology has been successfully used for protein engineering in mammalian cells. In this system, catalytically inactive dCas9 recruits variants of the cytidine deaminase base editor AID with MS2-modified sgRNAs to mutagenize endogenous targets (Hess et al, 2016). Using a similar strategy, Cas9 fused with base editors and sgRNAs targeting the coding sequence of OsALS were used to produce variants tolerant to the herbicide bispyribac sodium (BS) (Kuang et al, 2020). However, this system can only target short segments of DNA, and not in a continuous fashion, which may be desired for the synthetic evolution of traits. In the EvolvR system, which has thus far been used only in bacteria, CRISPR-guided nCas9 fused with nick-translating error-prone DNA polymerase nicks the target locus and performs mutagenesis at the target site (Halperin et al, 2018). Using CRISPR-X and EvolvR, high rates of mutagenesis at ~50-bp regions adjacent to the sgRNA target site have been achieved, but the mutation rate dropped with increasing distance from the sgRNA target window.

CRISPR-directed evolution (CDE) was recently performed by transforming rice (Oryza sativa) callus with CRISPR/Cas9 along with a pool of sgRNAs targeting the splicing factor locus OsSF3B1. Mutants were produced due to non-homologous end joining (NHEJ) under selection pressure imposed by the splicing inhibitor GEX1A. The recovered SF3B1 variants showed different levels of tolerance of GEX1A (Butt et al, 2019a, 2020b, 2021). Therefore, efficient mutagenesis requires dozens of sgRNAs to diversify the target gene. In addition, these CRISPR/Cas-based mutagenesis platforms have limited utility because of PAM sequence restrictions, the change of PAM sites by mutation, the change of sgRNA-binding sites by mutation, and the narrow genomic window adjacent to the sgRNA binding site, thus exhibiting an overall lack of efficiency for self-recurring continuous mutagenesis.

For continuous in vivo mutagenesis of target sequences several kilobases long, base editors can be fused with bacteriophage T7 RNA polymerase (T7 RNAP) (Moore et al, 2018; Chen et al, 2020; Park & Kim, 2021). The target sequences are then placed under the control of the T7 RNAP promoter (pT7). Mutations are introduced by fusion proteins of the base editor with T7 RNAP, which recognizes the pT7 and reads through the target DNA. Chimeric T7 RNAP deaminase enzymes can perform continuous mutagenesis of several kilobases of DNA and have been used for directed evolution, including MutaT7 in bacteria (Moore et al, 2018; Álvarez et al, 2020; Park & Kim, 2021), TRIDENT in yeast (Cravens et al, 2021), and TRACE in mammalian cells (Chen et al, 2020). However, the use of this T7 RNAP deaminase editing system to introduce localized sequence diversification, leading to synthetic evolution, has not been demonstrated in plants.

Here, we established a T7 RNAP deaminase editor system for targeted mutagenesis for the first time in plant cells. In transient experiments, we targeted the GFP sequence and achieved localized sequence diversification at high efficiency. Most of the transitions were C-to-T substitutions. Single-vector and two-vector approaches for the evolution of OsALS allowed us to recover gene variants conferring herbicide resistance, a trait of interest in rice. We identified herbicide-responsive residues in the resulting acetolactate synthase (ALS) protein, and generated ALS variants conferring herbicide resistance. This technique opens up myriad possibilities for synthetic evolution in plants, including the development of crops with increased resistance to changing climate conditions, resistance to pests and pathogens, and improved productivity.

# Results

### Design, construction, and testing of the T7 RNAP–DNA base editor in *Nicotiana benthamiana* transient assays

Bacteriophage T7 RNAP transcribes DNA sequences under the control of the T7 promoter. A fusion protein of T7 RNAP with a cytidine deaminase (Editor) could continuously edit the DNA bases downstream of the T7 promoter (Target) (Fig 1A). Once the cytidine deaminase (AID) converts C > U, the uracil–guanine (U–G) mismatch can be misread, resulting in a C > T or G > A transition. Alternatively,

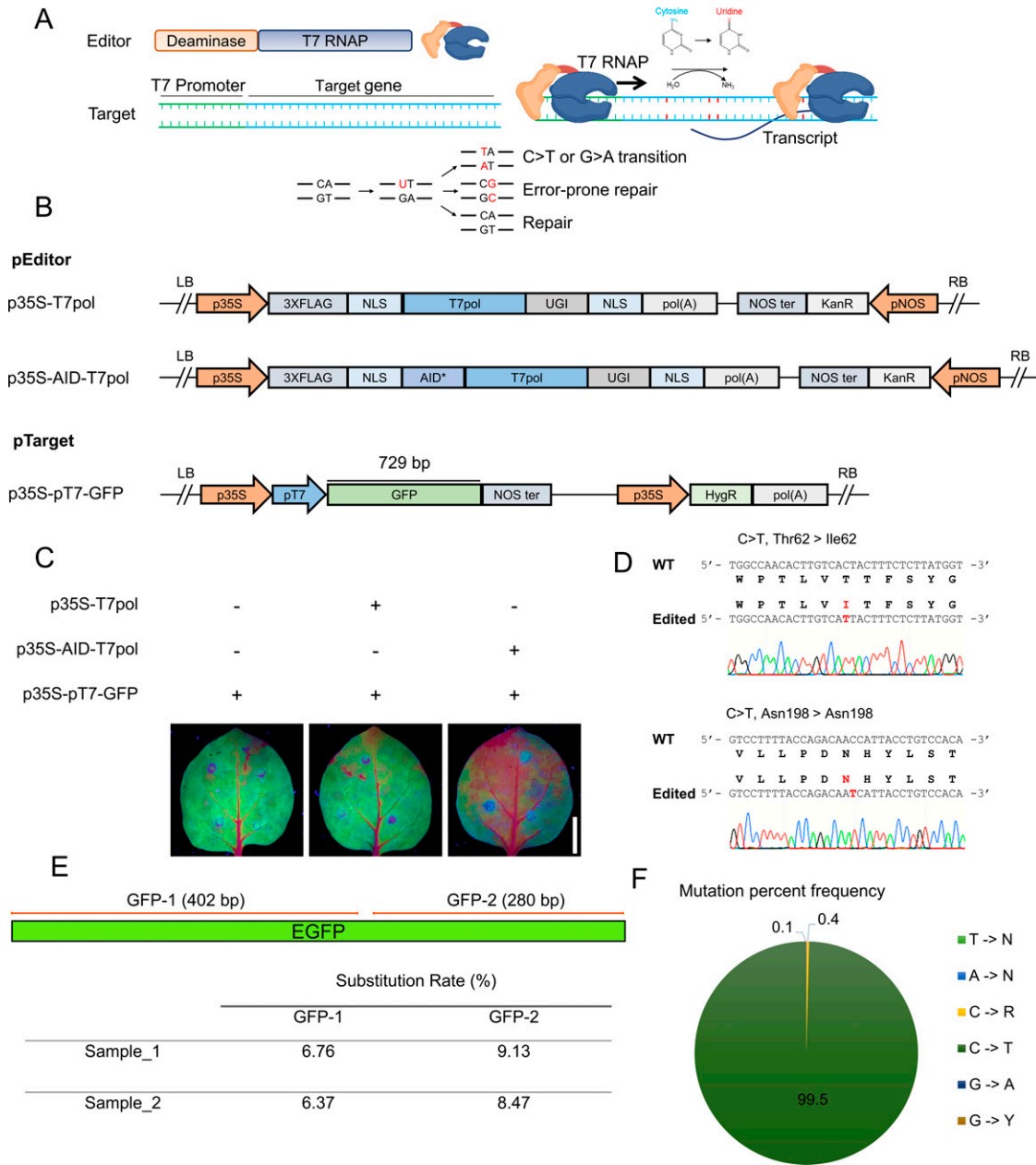

**Figure 1: A cytidine deaminase T7 RNA polymerase fusion mediates editing in *Nicotiana benthamiana* transient assays.**
**(A)** Schematic diagram of the targeted mutagenesis system. The hyperactive cytidine deaminase AID is fused to the N-terminus of T7 RNA polymerase (Editor) and recognizes the T7 promoter inserted upstream of the target gene (Target). Once the T7 polymerase transcribes the target gene, ssDNA is available for AID deaminase activity, which converts cytosine to uracil. The U-G mismatch can be misread, resulting in C > T or G > A substitutions. Alternatively, error-prone polymerase can be recruited through the mismatch-repair pathway, generating transitions and transversions near the lesion. **(B)** Design of the pEditor and pTarget plasmids used for transient expression in *N. benthamiana* leaves. pEditor without AID was used as a control for editing. **(C)** The leaves were Agro-infiltrated with pEditor and pTarget plasmids. At 3 dpi, images were taken under UV light to detect GFP. **(D)** Sanger sequencing analysis of p35S-GFP targeted by AID-T7 RNAP showing C > T substitutions. **(E)** Amplicon deep sequencing analysis of *GFP* targeted by p35S-AID-T7 RNAP; p35S-GFP co-infiltrated with p35S-T7 RNAP was used as the mock control. Two independently infiltrated leaf samples were analyzed. The PCR products GFP-1 and GFP-2 were processed using an Illumina kit, and samples were sequenced on the NovaSeq platform. The p35S-T7 RNAP data were used as a control and subtracted during analysis to detect the true substitutions rate during base editing. **(F)** Mutation percent frequency represents the distribution of all types of base edits found in *GFP* samples treated with p35S-AID-T7 RNAP. The data were analyzed using Geneious Prime.
Source data are available for this figure.

an error-prone polymerase could be recruited through the mismatch-repair pathway, generating transitions and transversions near the lesion (Odegard & Schatz, 2006) (Fig 1A).

To test the ability of the T7 RNAP deaminase fusion to induce mutations within a target region, we constructed the AID-T7 RNAP-UGI fragment with hyperactivating mutations (P266L, G645A, and

Q744R) (Chen et al, 2020) in the plant expression vectors pEditor (Fig 1B). We cloned the pAID-T7pol fragment using unique restriction enzymes to generate a full-length sequence flanked by attL1 and attL2 recombination sites, a nuclear localization signal fused to the N- and C-termini of the protein, and a 3x-HA tag fusion at the N-terminus to facilitate protein detection. Then, AID-T7 RNAP-UGI was subcloned into the destination binary vector pK2GW7 via Gateway recombination reaction to generate the overexpression construct p35S-AID-T7pol driven by the Cauliflower mosaic virus (CaMV) *35S* promoter. The sequence encoding the cytidine deaminase AID was removed for the mock control experiments (Fig 1B).

We tested this editing system by performing transient assays in *N. benthamiana* leaves using *GFP* as a target. *GFP* was expressed under the control of the T7 promoter as a binding target of T7 RNAP, the pTarget. For continuous expression of *GFP*, the constitutive CaMV 35S promoter was inserted upstream of the T7 promoter (Fig 1B). The binary vectors p35S-T7pol, p35S-AID-T7pol, and p35S-pT7-GFP were independently transformed into *Agrobacterium tumefaciens* via electroporation and co-infiltrated into *N. benthamiana* leaves for transient expression (Fig 1C). We assessed the rate of random target mutagenesis at 3 d post infiltration (dpi) by visualizing the GFP signal in infiltrated leaves under ultraviolet light (Fig 1C).

Then we extracted DNA from infiltrated leaves and amplified the ~1,600-bp region across the T7 promoter. The amplified fragments were cloned into the pJET1.2 vector and subjected to Sanger sequencing. Of the 40 reads examined, two (5%) showed C > T substitutions of the *GFP* sequence (Fig 1D). The first C > T mutation changes the threonine (ACT) at the 62nd position to isoleucine (ATT), whereas the second C > T substitution is a silent mutation of asparagine (AAC to AAT) at the 198th position (Fig 1D). Importantly, we did not observe any substitutions in the region 800 bp upstream of the T7 promoter.

For in-depth analysis, we performed amplicon deep sequencing. We amplified and sequenced the amplicons from samples co-infiltrated with p35S-AID-T7pol and p35S-pT7-GFP. For the mock control, we sequenced amplicons from samples co-infiltrated with p35S-T7pol and p35S-pT7-GFP. Two different regions of the *GFP* sequence downstream of the T7 promoter were amplified (Fig 1E) and two biological replicates each were performed for the experimental and mock treatments. Illumina TruSeq DNA nano libraries were prepared and analyzed on the NovaSeq platform, analyzing the resulting data using the web tool CRISPR-sub (Hwang et al, 2020) and subtracting the mock-treated reads from the experimental sample reads to calculate editing efficiency.

The base editing efficiency for sample 1 was 6.76% for GFP amplicon-1 and 9.13% for GFP amplicon-2. Similarly, the base editing efficiency for sample 2 was 6.37% for GFP amplicon-1 and 8.47% for GFP amplicon-2 (Fig 1E). No substitutions were observed in the mock-treated samples (Fig S1). The indel frequency was <0.15% (Fig S1).

To calculate the distribution of all types of base edits, we analyzed all experimental samples treated with p35S-AID-T7pol using Geneious Prime. On average, in all treated samples, >99% of edited reads had C > T substitutions (Fig 1F). Only 0.4% of reads showed C > R substitutions, whereas 0.1% of reads showed G > A substitutions (Fig 1F). These results demonstrate that the T7 RNAP–driven CBE generated in this study can target and edit gene sequences in plant cells. In our experiment, over 99% of these mutations were C > T transitions.

## Localized sequence diversification of *OsALS* in transgenic rice plants

We next expanded the applicability of the T7 RNAP deaminase editor system to mutate the genes of interest in stable plants using the rice gene acetolactate synthase (*OsALS*, *LOC_Os02g30630*, 1,935 bp). To this end, we cloned *OsALS* under the control of the T7 promoter in various rice expression vectors as a pTarget (Fig 2A) using two approaches. In the first approach, the single-vector system, we generated a vector harboring pEditor and pTarget. To avoid continuous targeting of pTarget in the progeny and to inhibit any possible activities in *Agrobacteria*, the two-vector system was designed such that pEditor and pTarget were in two different vectors (Fig 2A).

For the single-vector system, the AID-T7 RNAP-UGI fragment was cloned via Gateway recombination reaction under the control of the *Ubiquitin* promoter and inserted *OsALS* into this vector using unique restriction enzymes to generate the full-length vector p35S-pT7-ALS/pUBI-AID-T7pol (Fig 2A). The AID-T7pol fragment was removed from this vector and p35S-pT7-ALS only was used as the pTarget.

For the two-vector system, we also used the *N. benthamiana* expression vector p35S-AID-T7pol in rice. Rice callus was co-transformed with p35S-pT7-ALS and p35S-AID-T7pol via *Agrobacterium*-mediated transformation and a combination of hygromycin and G418 was used for callus selection and shoot regeneration; for the single-vector system, only hygromycin was used for selection and regeneration. Once transgenic plantlets were established in soil, we extracted the DNA and amplified the 2.2-kb fragment covering the complete *OsALS* sequence along with vector sequences. The amplified fragments were cloned into the pJET1.2 cloning vector and 10 reads were analyzed for mutagenesis assessment via Sanger sequencing (Fig 2B and C).

In the single-vector experiment, six of eight (75%) plants showed at least one read with a base substitution, whereas in the two-vector experiment, three of eight (37.5%) plants contained at least one read with base substitutions. For the single-vector approach, 2 of 10 reads in plant #1 contained C > T substitutions. These substitutions convert alanine 334 to valine (GCC to GTC) and histidine 415 to tyrosine (CAC to TAC) (Fig 2B). Plant #2 contained G > A and A > G substitutions, which convert glycine 133 to aspartic acid (GGC to GAC). The other mutation in plant #2 was a silent mutation at proline 248 (CCA to CCG) (Fig 2B). For the two-vector system, plant #1 contained two reads with G > A edits. These edits convert alanine 467 to threonine (GCA to ACA) and glycine 485 to aspartic acid (GCG to ACG) (Fig 2C). Plant #2 of the two-vector system contained two silent mutations in the same read: glutamine 102 (CAG to CAA) and serine 132 (TCC to TCT) (Fig 2C).

To perform detailed analysis of the substitution rate and to calculate the mutation percent frequency, we performed amplicon deep sequencing using the PacBio platform. We analyzed the data using the offline version of CRISPR-sub and subtracted p35S-pT7-ALS reads to calculate the substitution efficiency. Similar to the Sanger sequencing results, the substitution frequency was 1.06% and 0.53% for the single- and two-vector systems, respectively (Fig 2D). We also analyzed the frequency of each type of substitution using Geneious prime (Fig 2E). Interestingly, 36.1% of the mutations were G > A substitutions, 32.3% were C > T substitutions, 17.2% were T > N substitutions, 6.2% were G > Y substitutions, 6.1% were C > R

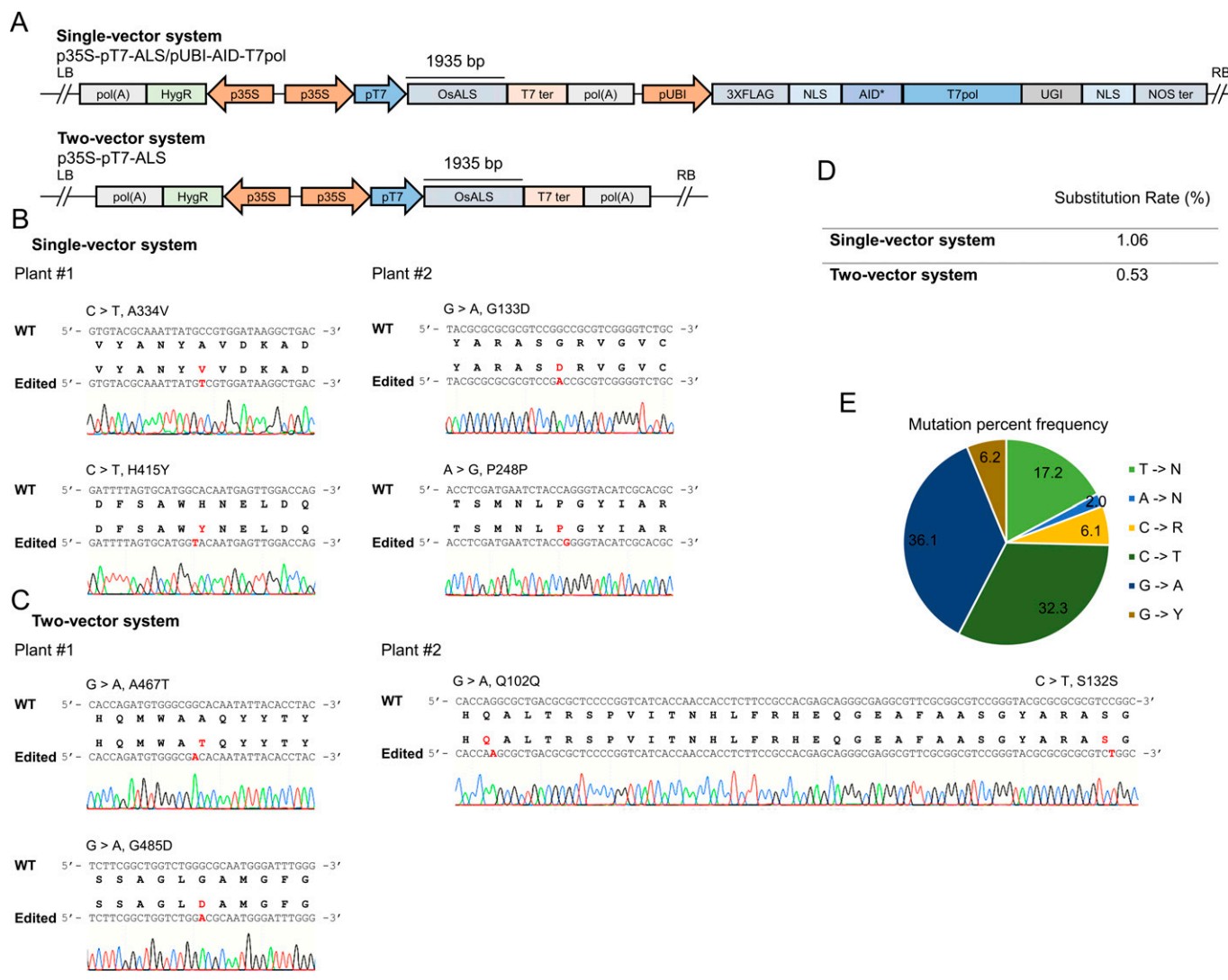

**Figure 2. Targeted base editing in stable rice lines.**
**(A)** Design of the plasmid used for Agrobacterium-mediated transformation of callus. In single-vector editing, the target and editor were cloned in the same vector and transformed into rice. In the two-vector system, *OsALS* was inserted into the vector under the control of the T7 promoter and co-transformed with pEditor. **(B)** Sanger sequencing analysis of regenerated rice shoots transformed with the single-vector or two-vector system. **(C)** 10 reads were analyzed per plant. Most seedlings showed C > T and G > A substitutions. **(D)** Amplicon deep sequencing analysis of *acetolactate synthase* (*ALS*) targeted with the single-vector or two-vector system. p35S:ALS without the base editor was used as a mock control. The amplicons were sequenced using PacBio technology. The substitution rate was calculated after subtracting the mock-treated sample reads. **(E)** Mutation percent frequency represents the distribution of base edits of all types found in the *ALS* samples with p35S-AID-T7 RNAP. The data are the means of the results for the single- and two-vector systems.
Source data are available for this figure.

substitutions, and 2.0% were A > N substitutions (Fig 2E). Overall, these results indicate that T7 RNAP–mediated targeted mutagenesis can be performed in transgenic rice plants, albeit at low but sufficient frequencies.

## Directed evolution of *OsALS* to identify herbicide-resistant mutations

To use and exploit the T7 RNAP deaminase editing system for trait engineering in plants, we use the rice gene *OsALS* for evolution of herbicide resistance. For this purpose, we performed *Agrobacterium*-mediated stable transformation of rice callus using our

single-vector and two-vector systems harboring *OsALS* as the target locus. After co-cultivation, the callus was transferred to selection medium supplemented with 0.5 or 0.75 µM BS to inhibit the growth of wild-type callus and exert selection pressure to generate *OsALS* variants. For the mock control, the callus was grown on selection medium without BS. After 2 wk of selection, five independent actively growing callus pieces were pooled and used to amplify the *OsALS* sequence in the transgene (Fig 3A). The full-length amplicon was subjected to deep sequencing using PacBio sequencing technology and the data analyzed using the offline version of CRISPR-Sub. To identify the BS-responsive regions of *OsALS*, we compared the reads from BS-treated samples with data from mock-

A

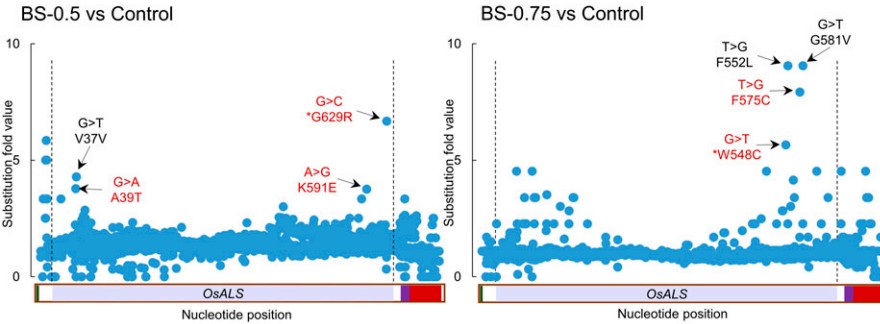

**Figure 3. Identification of bispyribac sodium (BS)-responsive regions at the *OsALS* locus.**
**(A)** The *OsALS* amplicon fragment in the T-DNA is shown. The arrows represent the primers used to amplify the *acetolactate synthase* sequence from the transgene. **(B)** The callus was transformed using the single-vector or two-vector system. After 2 wk of selection, five proliferated callus pieces on selection medium containing 0.5 and/or 0.75 μM BS were selected and pooled into a single sample. Five proliferated callus pieces on selection medium without BS were pooled into a single sample and used as a control. The data were analyzed by comparing the BS-treated versus control samples. The substitutions in the red cause amino acid changes, whereas in black are silent mutations.
Source data are available for this figure.

B

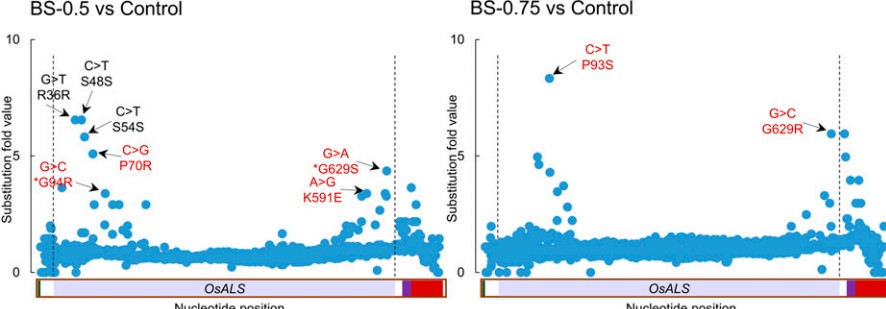

treated samples. The mutations that were common among BS-treated and mock-treated reads were subtracted during analysis.

Many more substitution mutations were observed at this locus in BS- versus mock-treated samples (Fig 3B). We considered these to be BS-responsive substitutions. Interestingly, these substitutions were enriched in the C- and N-terminal regions of the ALS protein. In samples transformed with the single-vector system and selected on 0.5 μM BS, we identified the enrichment of G > C causing the G629R substitution, G > A causing the A39T substitution, and A > G causing the K591E substitution. By contrast, in single-vector samples selected on 0.75 μM BS, we identified the enrichment of G > T causing the W548C substitution, and T > G causing F575C substitution (Fig 3B).

For samples transformed with the two-vector system and selected on 0.5 μM BS, we identified the enrichment of C > G causing the P70R substitution, G > C causing the G94R substitution, C > G causing the P70R substitution, A > G causing the K591E substitution, and G > A causing G629S substitution. By contrast, for two-vector samples selected on 0.75 μM BS, we identified the enrichment of C > T causing the P93S substitution and G > C causing the G629R substitution (Fig 3B). Interestingly, G629 appeared to be an important residue in our analysis, as substitutions at this site were enriched in samples transformed with the single-vector system

selected on 0.5 μM BS and in samples transformed with the two-vector system on both concentration of BS (Fig 3B). These results highlight the important residues in the *OsALS* sequence that are enriched in response to BS treatment.

### ALS mutant variants confer variable levels of herbicide resistance

To further engineer the BS-responsive regions at the *ALS* sequence and to generate herbicide-resistant *ALS* variants, we mutate the *OsALS* of pTarget, p35S:ALS, via site-directed mutagenesis (Fig 4A). We selected nine different herbicide-responsive point mutations (Fig 3) that cause substitutions A39T, P70R, P93S, G94R, W548C, F575C, K591E, G629R, and G629S (Fig 4A). The vectors with these point mutations were used to transform embryonic rice callus to generate transgenic rice plants harboring ALS mutant variants.

To validate the herbicide resistance of ALS mutant variants, we tested the progeny of these transgenic plants. We conducted germination and root inhibition assays at several BS concentrations (Figs S2 and S3). The seeds of wild-type and progeny of p35S:ALS without ALS mutation were used as controls. We observed that the germination of WT seeds was severely inhibited at 0.75 μM BS, whereas the germination of p35S:ALS seed with WT-ALS sequence was affected at 0.75 μM BS and severely inhibited at 1 μM BS (Figs 4B

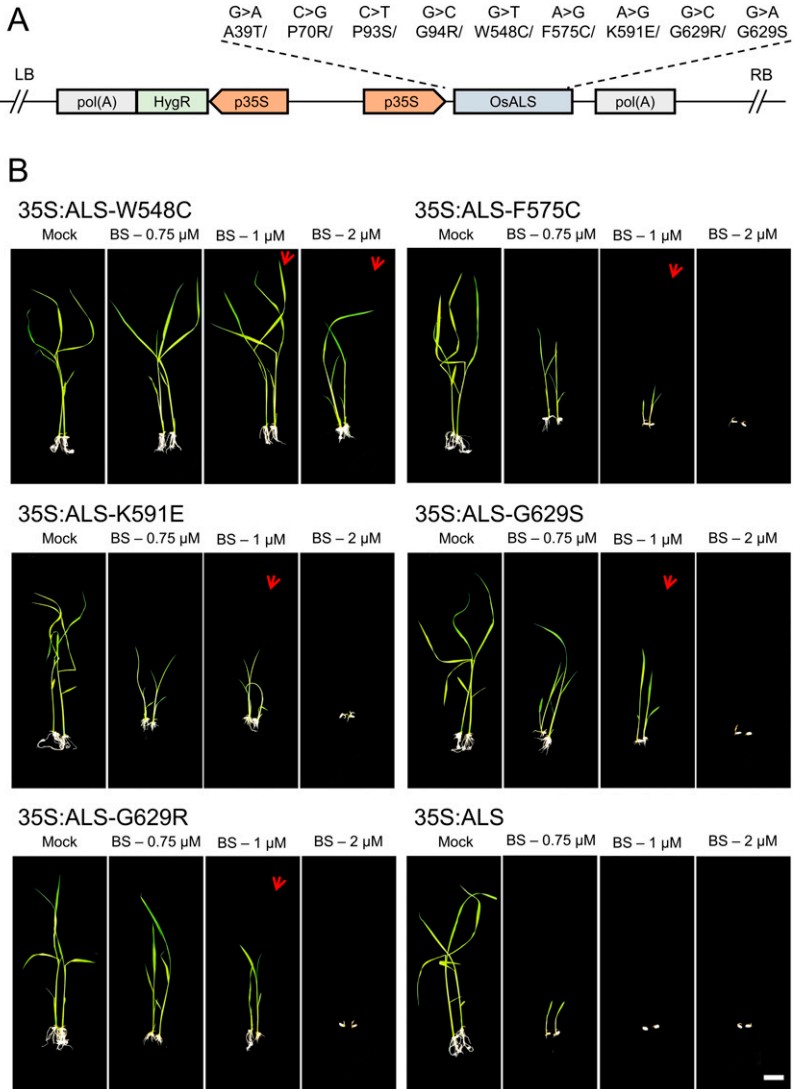

**Figure 4. Herbicide resistance analysis of transgenic rice plants expressing point mutations in the *acetolactate synthase* (ALS).**

**(A)** Nine mutant variants are generated for *OsALS*. Point mutations are introduced into the *OsALS* sequence using site-directed mutagenesis. *Agrobacterium*-mediated transformation was conducted of each of the p35S:ALS mutant variant. **(B)** Dose–response effects of bispyribac sodium (BS) treatment on germination of 35S:ALS-W548C, 35S:ALS-F575C, 35S:ALS-K591E, 35S:ALS-G629S, and 35S:ALS-G629R. Seeds of 35S:ALS with *OsALS* wild-type sequence was used as a control. Rice seeds were sterilized and germinated on ½ MS basal salt media plates for 2-wk with different concentrations of BS. Germination of 35S:ALS seeds was severely affected at 0.75 μM BS and completely inhibited at 1 μM BS. Germination of ALS mutant variants is less affected by 1 μM BS treatment. 35S:ALS-W548C germination is slightly affected even at 2 μM BS. Red arrows indicate the BS-resistant shoots. Three independent transgenic lines were tested for each of the ALS mutant variant.

and S2). We considered the 1 μM BS for screening of ALS mutant variants. We observed that the germination of ALS-W548C is insensitive to 1 μM BS and slightly affected at 2 μM BS (Fig 4B). The G629S and G629R are insensitive to 0.75 μM BS and showing resistance at 1 μM BS (Fig 4B). The F575C and K591E are resistant to BS at 0.75 and 1 μM concentration, however at lower levels than W548C, G629S, and G629R (Fig 4B). The other ALS mutant variants A39T, P70R, P93S, and G94R did not show resistance to BS like the p35S:ALS-WT control (Fig S2). We also tested the response of the seedlings of ALS mutant variants to different concentrations of BS (Fig S3). The BS resistance of seedlings is similar to our germination analysis, where W548C was showing the highest levels of resistance at 1.5 and 2 μM BS concentrations (Fig S3). The G629S and G629R are resistant to 1.5 μM BS and slightly affected at 2 μM BS (Fig S3). The F575C and K591E are showing slight resistant to BS at 1.5 μM and affected 2 μM concentration (Fig S3). All the other ALS mutant variants A39T, P70R, P93S, and G94R, including WT and p35S:ALS-WT control are sensitive to at 1.5 μM BS (Fig S3). Some of these ALS sites and the nearby sites G95,

W548, and G629 (Endo et al, 2007; Okuzaki et al, 2007; Butt et al, 2017; Zhang et al, 2020) are known to confer resistance to BS. However, we identified the novel ALS BS-resistant substitutions W548C and G629R, and residues F575 and K591, which were not reported before.

In conclusion, we successfully used the T7 RNAP deaminase editing system to identify the BS-responsive regions in the ALS sequence and validate for herbicide resistance trait engineering in rice.

## Discussion

Synthetic directed evolution involves targeted gene diversification, selection, and replication to obtain new functional biomolecules for use in research, biotechnology, and medicine (Zeymer & Hilvert, 2018; Simon et al, 2019; Morrison et al, 2020). The ideal synthetic directed evolution system must be performed in vivo in a continuous manner, preferably in the organism in which new variants are

desired (Gionfriddo et al, 2019; Butt et al, 2020b; Hendel & Shoulders, 2021). Directed evolution is extremely difficult in plants because of the nature of their anatomy and physiology; therefore, only a few successful attempts have been reported in plants based on CRISPR/Cas systems (Butt et al, 2019a; Kuang et al, 2020; Li et al, 2020). To date, no continuous directed evolution system has been established to evolve plant proteins within plant cells.

We previously demonstrated that directed evolution is possible in plants using our newly established CDE platform. A library of sgRNAs was synthesized to tile all of the available PAM sites of the spliceosomal component SF3B1 in rice (Butt et al, 2019a, 2021). The splicing inhibitor GEX1A was used as selection pressure to recover *OsSF3B1* variants with increased resistance to splicing inhibitors. However, the CDE platform has several drawbacks, including a tight, finite genomic window to the sgRNA target site and the lack of continuous mutagenesis. Here, in this study, we exploited the T7 RNAP deaminase editing system to overcome these limitations. In the current study, we establish an efficient, continuous, localized targeted mutagenesis platform that could be used for synthetic evolution in plants. We used a chimeric protein of a cytidine deaminase and T7 RNA polymerase (T7 RNAP) for targeted mutagenesis of genes controlled by the T7 promoter. We achieved high C > T editing efficiency in *N. benthamiana* transient assays, establishing the utility of the system in plants.

We further modified the system to create single- and two-vector systems and used them to generate stably transformed rice plants. The advantage of the two-vector system is that the pEditor is removed after segregation and only pTarget persists in the progeny. Unlike the transient assays in *N. benthamiana*, we observed not only C > T but also G > A transitions in these stable rice lines. We took advantage of this system to improve the herbicide resistance trait in rice. We applied selection pressure using an herbicide and identified the herbicide-responsive residues in the *OsALS* gene sequence via a comparison with mock treatment. Ideally, such modifications could be introduced into the rice genome via HDR-based technologies such as prime editing, RNA-templated DNA repair, or Cas9-VirD2 fusion-mediated repair (Butt et al, 2017, 2020a; Anzalone et al, 2019; Ali et al, 2020). The evolution of *ALS* performed in this study represents a proof of concept; this system could be applied to other proteins of interest.

T7 RNAP–based DNA editor system offers several advantages for the directed evolution of proteins in plant. Foremost, the plant proteins are evolved within the signaling nexus of the plant cell. The timing, location, and kinetics of these interactions are critical for the proper function of the proteins. The protein folding, post-translational modifications, localizations and interactions with other proteins are needed for performance in the cell (Hendel & Shoulders, 2021).

Second, it ousted the designing and cloning of sgNRA library to tile the target gene for saturation mutagenesis. Thus, it edits the target bases without PAM restrictions. For crop genes for which the functional SNPs are not known, T7 RNAP–based DNA editor system can be used to produce valuable genetics variants with improved agronomic performances.

Third, the T7 RNAP–based DNA editor system constantly mutates the target gene leading to increased genetic diversity. It enables the continuous directed evolution of the target gene under the varying selection pressure conditions.

Last but not the least, the mutation process can be fined tuned by controlling the expression of the T7 RNAP base editor. Then the selection pressure can be applied during any stage of plant development keeping the T7 RNAP base editor under the control of an inducible promoter.

We observed different mutation ratios for transient expression of GFP in tobacco and stable expression of *OsALS* in rice. In the transient assays of the tobacco leaves, the DNA is available on the episome, whereas in the stable rice transformation, the DNA is integrated into the chromosomes. The episome lacks the physiological chromatin; thus, the chromatin modifications, the binding of chromatin modifiers, transcription factors and cofactors, or chromatin accessibility are different factors which may impact the mutation rate among transient and stable assays (Inoue et al, 2017).

Nevertheless, this system could be further amended by using other CBEs, ABEs, or dual base editors containing ABEs and CBEs to diversify the editing. The editing efficiency could also be tuned by using repair factors, as described for yeast (Cravens et al, 2021). The continuous evolution of the entire target sequence is sometimes undesirable, as the editing of the conserved regions of a gene should sometimes be avoided. This problem could be overcome by using dCas9 as a "road block" and limiting the activity of T7 RNAP at the sgRNA binding site (Álvarez et al, 2020). Another approach is to create a dual T7 promoter/terminator for the target, where the second T7 promoter/terminator pair transcribes the target gene in the reverse direction (Moore et al, 2018; Park & Kim, 2021). The system can be further applied to pathogen- or stress-responsive genes where the SNPs of the mutant variants will be identified in the target gene by comparing the stress treatment with the mock treatment. These stress-responsive SNPs could then be introduced into the genome using an efficient HDR editing system.

Our study has exploited the T7 RNAP base editor for continuous and localized gene sequence diversification for the first time in the plant system. We successfully developed and used the T7 RNAP base editor for localized sequence diversification of *GFP* in *Nicotiana benthamiana*. This localized sequence diversification modality will unlock the power of synthetic evolution in plants. The T7 RNAP system could be used for localized sequence diversification and evolution of variants of interest, conferring a specific trait like herbicide resistance at the callus stage. Once these mutant variants have been identified, they can be engineered via CRISPR-Cas systems to generate transgene-free and foreign DNA-free mutant variants conferring the trait of interest. We have shown the power of this system to identify variants of *OsALS* under the selection pressure of herbicide and developed herbicide resistance in rice. This study opens up many possibilities for synthetic evolution in plant species, including developing crop plants resilient to climate change conditions, resistant to pests and pathogens, and improved productivity.

# Materials and Methods

### Plant materials

*O. sativa* L. ssp. *japonica* cv. Nipponbare was used for the rice experiments. Wild-type *N. benthamiana* plants were used for the

*N. benthamiana* experiments. All wild-type and transgenic plants were grown in a greenhouse at 28°C/25°C day/night temperatures under a natural light/dark cycle.

## Vector construction and cloning strategy

The pEditor fragment of AID_T7_UGI was amplified from the vector pcDNA3.1 (+)-Hyperactive AID-T7 RNA Polymerase-UGI-T2A-td Tamato (138610; Addgene; [Chen et al, 2020]) using primers SphI_AID_T7_F and NSiI_XbaI_AID_T7_R, digested with SphI and NsiI, and cloned into attL1_3xFLAG_attL2_pUC57 to generate attL1_3xFLAG_AID_T7_UGI _attL2_pUC57.

For the *N. benthamiana* experiments, the 3xFLAG_AID_T7_UGI fragment was cloned into pK2GW7 via LR reaction under the control of the 35S promoter to generate pEditor p35S_3xFLAG_AID_T7_UGI _pK2GW7. To prepare p35S_3xFLAG_T7_UGI_pK2GW7 without AID, the p35S_3xFLAG_AID_T7_UGI _pK2GW7 vector was digested with XmaI and re-ligated. For GFP expression (pTarget), pLSLGFP was used (Addgene-51494; [Baltes et al, 2014]), and SIR and the downstream LIR were removed by digestion with SacI. To clone the T7 promoter upstream of *GFP* for expression in *N. benthamiana*, oligos with the T7 promoter and RBS sequences were annealed with SbfI and XbaI overhangs and cloned into pL-GFP_SacI by replacing the upstream LIR. To clone the 35S promoter upstream of the T7 promoter, oligos (35sP_HinD_F and 35sP_PstI_HinD_R) were designed to amplify the 35S promoter. The 35S promoter was cloned upstream of the T7 promoter using HindIII and PstI to generate the final plasmid p35S_pT7_GFP_NOSterm.

For the single-vector system in rice, the fragment 3xFLA-G_AID_T7_UGI was cloned into pRGEB32 via LR reaction under the control of the *OsUbiquitin* promoter to generate pEditor pUBI_3xFLAG_AID_T7_UGI_pRGEB32. Oligos for the T7-promoter_RBS_MCS_T7-terminator_CaMV poly(A) signal sequence were annealed and ligated with HindIII-SbfI overhangs and cloned into the rice pEditor vector. The 35S promoter was cloned upstream of the T7 promoter in the rice pEditor vector using only HindIII. The *OsALS* sequence (*LOC_Os02g30630*) was amplified with oligos ApaI_AvrII_ALS_F and XbaI_KpnI_ALS_R (1.963 Kb) and cloned under the control of pT7 or p35S-pT7 in rice pEditor vectors via digestion with ApaI and XbaI.

In rice for the two-vector system, pEditor and pTarget were transformed into rice as separate vectors. *N. benthamiana* pEditor p35S_3xFLAG_AID_T7_UGI _pK2GW7 was used as pEditor for rice. The AID_T7pol_UGI fragment was removed from p35S_pT7_ALS_AID_T7pol_UGI_pRGEB32 by BsrGI digestion to prepare p35S_pT7_ALS_pRGEB32 as pTarget. All the primers sequences used in this study are given in Table S1.

## Agro-infiltration of *N. benthamiana* leaves and GFP imaging

Constructs harboring pEditor (p35S_3xFLAG_AID_T7_UGI_pK2GW7), pEditor without AID (p35S_3xFLAG_T7_UGI_pK2GW7), pTarget (p35S_pT7_GFP_NOSterm_T7term), and P19 were individually electroporated into *Agrobacterium tumefaciens* strain GV3101. Single colonies grown for two nights in selective medium were centrifuged, resuspended in infiltration medium (10 mM MES, pH 5.7, 10 mM CaCl$_2$, and 150 $\mu$M acetosyringone), and incubated at ambient temperature for 2 h. For infiltration into wild-type plants, the

cultures were mixed at an OD600 ratio of 0.2:0.4:0.4 (P19, pEditor, pTarget) and infiltrated into 3- to 4-wk-old leaves of *N. benthamiana* plants with a 1-ml needleless syringe. GFP expression was observed at 3, 5, and 7 dpi using a hand-held UV light. Photographs were taken with a Nikon camera under UV light.

## Rice transformation and mutant screening

Rice transformation plasmids were introduced into *A. tumefaciens* strain EHA105. Agrobacterium-mediated rice transformation was performed as described previously (Hiei & Komari, 2008; Butt et al, 2019b). Calli transformed with pRGEB32 were selected on medium containing 50 mg/l hygromycin B. Calli transformed with pK2GW7 were selected on medium containing 150 mg/l G418 and regenerated on 100 mg/l G418. To select resistant variants of pTarget *OsALS*, BS was used at a concentration of 0.5 or 0.75 $\mu$M.

After 1 wk of growth, when plants were established in soil, DNA was extracted from a leaf sample. To genotype pTarget *OsALS*, the HindIII_T7_PT_F and Sbf_T7_PT_R oligos were used to amplify the *OsALS* sequence from T-DNA (2.28 Kb) and cloned using a CloneJET PCR Cloning Kit (K1231). The same fragment was sequenced with three to four overlapping primers to cover the entire coding sequence. At least 10 colonies were subjected to Sanger sequencing to analyze the mutation.

## Amplicon deep sequencing

For amplicon deep sequencing of *GFP*, two regions were amplified with the primers GFP_F7 + GFP_R1 (402 bp) and PGD_GFP_F+ GFP_R2 (280 bp). Purified PCR products were used to prepare Illumina TruSeq DNA Nano libraries according to the manufacturer's instructions. The libraries were analyzed on the NovaSeq platform. The online tool CRISPR-Sub was used for data analysis (Hwang et al, 2020). pTarget-GFP samples co-infiltered into *N. benthamiana* leaves with p35S_3xFLAG_T7_UGI_pK2GW7 without AID were used as the control during data analysis. The edited samples were compared with the control samples to calculate the substitution rate.

For amplicon deep sequencing of rice callus samples, five independent callus pieces per treatment were collected after 2 wk of culture on medium without BS, with 0.5 $\mu$M BS, or with 0.75 $\mu$M BS. Equal amounts of samples from the same treatment groups were pooled together and used for DNA extraction. The *ALS* sequence from T-DNA was amplified using the primers HindIII_T7_PT_F and Sbf_T7_PT_R (2,278 bp). The purified PCR products were sequenced using PacBio Sequel I technology. The data were analyzed using the offline version of CRISPR-Sub. To avoid the factor of sequencing errors and somaclonal variations we subtracted the mock-treated sample (without AID) reads from the experimental sample (with AID) reads to calculate editing efficiency.

Data from the 0.5 $\mu$M BS- and 0.75 $\mu$M BS-treated samples were compared with data from untreated samples to identify the BS-responsive substitutions.

Geneious Prime software was used to calculate the mutation percent frequency of each type of base editing for the NovaSeq and PacBio data. The results of the data analysis are available as Additional_File_2 for Mutation rate GFP and ALS, Additional_File_3

for Substitution rate for ALS and Additional_File_4 for BS-responsive regions at the ALS.

### Generation of transgenic ALS mutant variants

The herbicide-responsive point mutations were introduced into the ALS sequence of the vector pTarget, p35S_pT7_ALS_pRGEB32, via site-directed mutagenesis (GenScript). The vectors with ALS mutations were transformed into wild-type rice callus and selected on medium containing 50 mg/l hygromycin B. The seeds of the progeny plants were used for herbicide resistance analysis.

### BS-resistance analysis of ALS mutant variants

For seedling analysis, the seeds were de-husked, sterilized, and grown vertically on square plates containing ½ MS media (with 50 mg/l hygromycin B). Seeds of wild-type and pTarget, p35S_pT7_ALS_pRGEB32, without ALS mutations were used as controls. After 3 d, seedlings with similar root growth were transferred to ½ MS plates supplemented with BS. The root tips were marked to observe growth. The seedlings were grown vertically for another 14-d and then imaged.

For germination analysis, the seeds were de-husked, sterilized, and grown vertically on square plates containing ½ MS media containing different concentrations of BS. Seeds of wild-type and pTarget, p35S_pT7_ALS_pRGEB32, without ALS mutations were used as controls. The seeds were grown vertically for 14-d and then imaged.

## Supplementary Information

## Acknowledgements

We would like to thank members of the genome engineering and synthetic biology laboratory at KAUST for their critical discussion and technical help in this work. This work was funded by KAUST-baseline funding to M Mahfouz.

### Author Contributions

H Butt: data curation, investigation, visualization, methodology, and writing—original draft.
JLM Ramirez: investigation and methodology.
M Mahfouz: conceptualization, supervision, funding acquisition, investigation, visualization, and writing—original draft, review, and editing.

### Conflict of Interest Statement

The authors declare that they have no conflict of interest.

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
