## [Reviewer comments · Life Science Alliance]

Life Science Alliance

Synthetic evolution of herbicide resistance using a T7 RNAP-based random DNA base editor

Haroon Butt, Jose Ramirez, and Magdy Mahfouz

DOI: <https://doi.org/10.26508/lsa.202201538>

Corresponding author(s): Magdy Mahfouz, King Abdullah University of Science and Technology

Review Timeline:

Submission Date:	2022-05-30
Editorial Decision:	2022-07-18
Appeal Requested:	2022-07-26
Appeal Granted:	2022-07-28
Editorial Decision:	2022-08-29
Revision Received:	2022-09-03
Editorial Decision:	2022-09-15
Revision Received:	2022-09-16
Accepted:	2022-09-19

Scientific Editor: Novella Guidi

Transaction Report:

July 18, 2022

Re: Life Science Alliance manuscript #LSA-2022-01538

Prof. Magdy M. Mahfouz
King Abdullah University of Science and Technology
4700 KAUST
Thuwal 23955
Saudi Arabia

Dear Dr. Mahfouz,

Thank you for submitting your manuscript entitled "Synthetic evolution of herbicide resistance using a T7 RNAP-based random DNA base editor" to Life Science Alliance. The manuscript has now been seen by expert reviewers, whose reports are appended below. Unfortunately, after an assessment of the reviewer feedback, our editorial decision is against publication in Life Science Alliance.

Given these Reviewer concerns, we are afraid that we are unable to proceed further with the paper. We are thus returning your manuscript to you at this time.

We are sorry our decision is not more positive, but hope that you find the reviews constructive. Of course, this decision does not imply any lack of interest in your work and we look forward to future submissions from your lab.

Thank you for your interest in Life Science Alliance.

Sincerely,

Reviewer #1 (Comments to the Authors (Required)):

The authors use a known system, but in plants. It consists of a T7 RNA Pol that makes errors if joined to a deaminase and then a target sequence that has a T7 binding site that will together drive mRNA will errors in them.

I was surprised a bacteriophage RNA polymerase makes suitable mRNA in plant cells?

Figure 1C, why if the error rate was low is most of the GFP-green lost when the 35S-expressed deaminase-coupled T7 pol is co-infiltrated with p35S-pT7-GFP? Shouldn't most of the GFP being made still be normal?

Line 207 "across the T7 promoter", this is a little confusing as the amplicon was more than the 800 bp of the T7 promoter, it also must have included the GFP ORF?

Very confusing for me is this: Why are DNA amplicons being amplified to evaluate the mutation frequency caused by a deaminase-RNA polymerase? The errors/mutations are being made at the level of mRNA presumably and leading to mutant proteins and yet the evaluation of mutations is being done by doing DNA PCR or DNA-seq (Illumina) deep sequencing? There's no mention of reverse transcription prior to amplicon generation by PCR or libraries coming from RNA present?

Line 219 I was surprised that after PCR (amplicon) and Illumina that no substitutions were seen as surely these two processes introduce bona fide mutations (PCR) and 'mutations' by Illumina read errors?

Line 290 What is the "hotspot" of the ALS sequence?

Line 311-313 I don't understand why site directed mutagenesis was done to make transgenic rice with known ALS mutations giving herbicide resistance? All that shows is you can make transgenic rice with ALS mutations?

All in all, I am sorry but there are too many aspects that confuse me in this manuscript, especially mutations introduced at the

RNA level being evaluated by sequencing DNA.

I have read Reviewer #2's review and none of my questions seem answered. Reviewer 2 only gives the authors 1 thing to do seeing it as an "excellent piece of work". I still am very confused by the work and do not support it as it is or until someone can explain/answer my comments.

Reviewer #2 (Comments to the Authors (Required)):

Directed evolution is a powerful engineering strategy for discovering nucleic acids or protein variants that have desired binding abilities or molecular functions. The manuscript by Haroon Butt et al. reports and describes an exciting strategy to enable high-efficiency localized sequence diversification, which is essential for unlocking the power of directed evolution applications in plant species. They used a T7 Polymerase fusion with deaminase for the first time in plants to enable random base editing. Recently, multiple papers reported random DNA base editor applications in other eukaryotic species that can cover an entire gene for targeted sequence diversification or continuous mutagenesis, making the method very attractive for targeted synthetic gene evolution in eukaryotic cells. These papers include Chen et al. 2020 Nature Biotechnology, Alvarez et al. 2021 Nature Communications, Cravens et al 2021 Nature Communications, etc. These papers reported a similar system that consists of a target cassette where a T7 polymerase base editor drives the target gene.

To my knowledge, it is the first attempt to use this strategy in plants. The data and results are quite exciting and convincing. Intriguingly, this study established a T7-base editor for targeted sequence diversification and continuous mutagenesis. The authors designed, built, and tested the activity of the T7 base editor and could show the high efficiency of base editing in plants.

Moreover, the authors showed random mutagenesis in a target gene by both transient and stable expression of T7 RNAP fused to AID and UGI in *Nicotiana benthamiana* and rice, respectively. They used NB to demonstrate the activity of the enzyme in transient assays. In rice, they attempted to evolve the ALS gene to confer resistance against BS herbicide through the T7 RNAP-base editor for sequence diversification and under the BS selection pressure. Intriguingly, they found candidate substitutions that can confer BS resistance in the OsALS gene using this approach. Thus, they demonstrated the application of T7 RNAP-based random mutagenesis in plants. For applications, the evolved variants can be engineered and introduced into the OsALS sequence in the genome via precision editing, including, for example, prime editing. I am quite pleased by the power of this modality and its ability to diversify plant genome sequences to increase variations and our ability to expand, expedite, and discover and develop new traits of value. This work is timely, very well designed, with compelling data and evidence supporting the work's conclusions.

For more clarity, in Figure 3B, the authors should describe the details of the red-colored substitution and asterisks in the legend. In conclusion, this is an excellent piece of work that merits its publication

Reviewer #1 (Comments to the Authors (Required)):

The authors use a known system, but in plants. It consists of a T7 RNA Pol that makes errors if joined to a deaminase and then a target sequence that has a T7 binding site that will together drive mRNA will errors in them.

Response: We thank the reviewer for raising these important questions. However, in our view, the reviewer is confused, failed to get the idea and did not understand this study. The AID deaminates the Cytosine to Uracil **in DNA not mRNA**. The DNA substrate is available to AID as a result of T7 RNAP activity.

As we have described in L176-L179 and Figure 1A, "Once the cytidine deaminase (AID, which converts C>U) is generated, the uracil–guanine (U–G) mismatch can be misread, resulting in a C>T or G>A transition. Alternatively, an error-prone polymerase could be recruited through the mismatch-repair pathway, generating transitions and transversions near the lesion (Figure 1A)."

We have convincingly addressed all the points as shown below.

1- I was surprised a bacteriophage RNA polymerase makes suitable mRNA in plant cells?

Response: The T7 RNAP guided base editing system is already used in prokaryotes and eukaryotes. As we described in the introduction L152-157 "Chimeric T7 RNAP–deaminase enzymes can perform continuous mutagenesis of several kilobases of DNA and have been used for directed evolution, including MutaT7 in bacteria (Alvarez et al 2020, Moore et al 2018, Park & Kim 2021), TRIDENT in yeast (Cravens et al 2021), and TRACE in mammalian cells (Chen et al 2020). However, the use of this T7 RNAP–deaminase editing system to introduce localized sequence diversification, leading to synthetic evolution, has not been demonstrated in plants."

For continuous expression of the target gene, we used the 35S promoter with T7 promoter.

2- Figure 1C, why if the error rate was low is most of the GFP-green lost when the 35S-expressed deaminase-coupled T7 pol is co-infiltrated with p35S-pT7-GFP? Shouldn't most of the GFP being made still be normal?

Response: We agree with the reviewer. We repeated the experiment more than three times and always observed the low GFP signal in the samples infiltrated with T7 RNAP-AID. This might be

due to the expression of T7 RNAP-AID which negatively affect the Agrobacterium growth. However, this is beyond the scope of this study.

3- Line 207 "across the T7 promoter", this is a little confusing as the amplicon was more than the 800 bp of the T7 promoter, it also must have included the GFP ORF?

Response: We thank the reviewer for this comment. We sequenced 800 bp upstream of the T7 promoter and 800 bp downstream of the T7 promoter. We changed the 800 to ~1600 bp in the revised version of the manuscript.

Very confusing for me is this: Why are DNA amplicons being amplified to evaluate the mutation frequency caused by a deaminase-RNA polymerase? The errors/mutations are being made at the level of mRNA presumably and leading to mutant proteins and yet the evaluation of mutations is being done by doing DNA PCR or DNA-seq (Illumina) deep sequencing? There's no mention of reverse transcription prior to amplicon generation by PCR or libraries coming from RNA present?

Response: As we described in the first paragraph of the rebuttal, the mutations will be introduced in the DNA not RNA. The T7 RNAP will facilitate to unzip the DNA and expose to AID for deamination.

Line 219 I was surprised that after PCR (amplicon) and Illumina that no substitutions were seen as surely these two processes introduce bona fide mutations (PCR) and 'mutations' by Illumina read errors?

Response: In our analysis, we used web tool CRISPR-sub, and subtracted the mock-treated samples from the experimental samples to calculate editing efficiency/mutation rate. If there are any bona fide mutations, those were also subtracted.

Line 290 What is the "hotspot" of the ALS sequence?

Response: We thank the reviewer for this comment. We used the term "hotspot" used by the authors of the CRISPR-sub software Hwang et al., 2020. We deleted this sentence from the revised manuscript to avoid any confusion and for smooth reading.

Line 311-313 I don't understand why site directed mutagenesis was done to make transgenic rice with known ALS mutations giving herbicide resistance? All that shows is you can make transgenic rice with ALS mutations?

Response: The substitutions and residues confirmed by producing transgenic rice. As we described in the L340-341, "However, we identified the novel ALS BS-resistant substitutions W548C and G629R, and residues F575 and K591, which were not reported before."

All in all, I am sorry but there are too many aspects that confuse me in this manuscript, especially mutations introduced at the RNA level being evaluated by sequencing DNA.

I have read Reviewer #2's review and none of my questions seem answered. Reviewer 2 only gives the authors 1 thing to do seeing it as an "excellent piece of work". I still am very confused by the work and do not support it as it is or until someone can explain/answer my comments.

Reviewer #2 (Comments to the Authors (Required)):

Directed evolution is a powerful engineering strategy for discovering nucleic acids or protein variants that have desired binding abilities or molecular functions. The manuscript by Haroon Butt et al. reports and describes an exciting strategy to enable high-efficiency localized sequence diversification, which is essential for unlocking the power of directed evolution applications in plant species. They used a T7 Polymerase fusion with deaminase for the first time in plants to enable random base editing. Recently, multiple papers reported random DNA base editor applications in other eukaryotic species that can cover an entire gene for targeted sequence diversification or continuous mutagenesis, making the method very attractive for targeted synthetic gene evolution in eukaryotic cells. These papers include Chen et al. 2020 Nature Biotechnology, Alvarez et al. 2021 Nature Communications, Cravens et al 2021 Nature Communications, etc. These papers reported a similar system that consists of a target cassette where a T7 polymerase base editor drives the target gene.

To my knowledge, it is the first attempt to use this strategy in plants. The data and results are quite exciting and convincing. Intriguingly, this study established a T7-base editor for targeted

sequence diversification and continuous mutagenesis. The authors designed, built, and tested the activity of the T7 base editor and could show the high efficiency of base editing in plants.

Moreover, the authors showed random mutagenesis in a target gene by both transient and stable expression of T7 RNAP fused to AID and UGI in *Nicotiana benthamiana* and rice, respectively. They used NB to demonstrate the activity of the enzyme in transient assays. In rice, they attempted to evolve the ALS gene to confer resistance against BS herbicide through the T7 RNAP-base editor for sequence diversification and under the BS selection pressure. Intriguingly, they found candidate substitutions that can confer BS resistance in the OsALS gene using this approach. Thus, they demonstrated the application of T7 RNAP-based random mutagenesis in plants. For applications, the evolved variants can be engineered and introduced into the OsALS sequence in the genome via precision editing, including, for example, prime editing. I am quite pleased by the power of this modality and its ability to diversify plant genome sequences to increase variations and our ability to expand, expedite, and discover and develop new traits of value. This work is timely, very well designed, with compelling data and evidence supporting the work's conclusions.

Response: We thank the reviewer for the very positive comments, compliments, and encouragement. We are deeply pleased with these comments.

For more clarity, in Figure 3B, the authors should describe the details of the red-colored substitution and asterisks in the legend.

Response: We thank the reviewer for this comment. We modified the figure legend and added the information in the revised version of this manuscript.

In conclusion, this is an excellent piece of work that merits its publication.

August 29, 2022

Re: Life Science Alliance manuscript #LSA-2022-01538R-A

Prof. Magdy M. Mahfouz
King Abdullah University of Science and Technology
4700 KAUST
Thuwal 23955
Saudi Arabia

Dear Dr. Mahfouz,

Thank you for submitting your manuscript entitled "Synthetic evolution of herbicide resistance using a T7 RNAP-based random DNA base editor" to Life Science Alliance. The manuscript was assessed by an expert reviewer, whose comments are appended to this letter. We invite you to submit a revised manuscript addressing the Reviewer comments.

Thank you for this interesting contribution to Life Science Alliance. We are looking forward to receiving your revised manuscript.

Sincerely,

B. MANUSCRIPT ORGANIZATION AND FORMATTING:

Reviewer #3 (Comments to the Authors (Required)):

In this manuscript, Butt et al., developed a previously reported T7 RNAP-based base editing system in plant. The authors also shown the application potential of this tool in plant by employing the BE for synthetic directed evolution of rice ALS gene to create the rice varieties with improved herbicide resistance. The manuscript is well-written and describes a new possibility in plant to artificially evolve the key genes to generate variants with improved traits, However, there are lots of places not described clearly or confusing.

1. In the line 129-130, "However, this system can only target short segments of DNA, and not in a continuous fashion" What do you mean by saying "not in a continuous fashion"? In my view, CRISPR-based BE also can continuously edit base.
2. I am very confused with Figure 1A, after deamination, C::G becomes U::G mismatch, which could probably convert into U::A after MMR if edited strand as the repair template. Why the bystander pair A::T could be converted into G::C by error-prone polymerase?
3. In the line 176-177, "Once the cytidine deaminase (AID, which converts) is generated, the uracil-guanine (U- G) mismatch can be misread". Is it "once the uracil is generated" ? Even though AID is generated, C>U is generated in a very low frequency. In addition, there seems no relationship between AID generation and U-G misread. Please modify the sentence to make it clearer.
4. In the line 147, the author described the T7 RNAP-deaminase tool "For continuous in vivo mutagenesis of target sequences several kilobases long", what is the maximum length for targeted gene with efficient base editing? Does the mutation rate decrease with increasing distance from T7 promoter?
5. When the authors tested T7 RNAP-deaminase in *Nicotiana benthamiana*, the most resulted mutations are C > T or G > A. However, when they applied the tool to evolve rice ALS, the most mutations generated are not them instead of other random conversions (figure 3 and figure 4). Is it likely that the mutations in ALS are caused by transcription error of T7 RNAP rather than deaminases?
6. If T7 RNAP-deaminase strategy can result in genome-wide DNA or RNA off-target like CRISPR-based BEs?
7. Since the T7 RNAP-deaminase strategy has been already reported in other organisms, what the differences from them when applied it in plant (e.g. efficiency, editing window)?
8. What are the drawbacks of this strategy at current stage compared with other synthetic directed evolution methods? How to improve in the future?

Reviewer #3 (Comments to the Authors (Required)):

In this manuscript, Butt et al., developed a previously reported T7 RNAP-based base editing system in plant. The authors also shown the application potential of this tool in plant by employing the BE for synthetic directed evolution of rice ALS gene to create the rice varieties with improved herbicide resistance. The manuscript is well-written and describes a new possibility in plant to artificially evolve the key genes to generate variants with improved traits. However, there are lots of places not described clearly or confusing.

Response: We thank the reviewer for the positive comments and compliments and for raising these essential questions. We have convincingly addressed all the points as shown below.

1. In the line 129-130, "However, this system can only target short segments of DNA, and not in a continuous fashion" What do you mean by saying "not in a continuous fashion"? In my view, CRISPR-based BE also can continuously edit base.

Response: We thank the reviewer for this comment. The CRISPR-based BE systems are discontinued once the bases are mutated in the PAM sites or sgRNA binding sites; hence the crRNA cannot target the same region again. In the following lines 143-146, we describe the drawbacks of the CRISPR-based BE. We added "the change of sgRNA binding sites by mutation" and modified the sentence in the revised version of the manuscript as below:

"In addition, these CRISPR/Cas-based mutagenesis platforms have limited utility due to PAM sequence restrictions, the change of PAM sites by mutation, the change of sgRNA binding sites by mutation, and the narrow genomic window adjacent to the sgRNA binding site, thus exhibiting an overall lack of efficiency for self-recurring continuous mutagenesis."

2. I am very confused with Figure 1A, after deamination, C::G becomes U::G mismatch, which could probably convert into U::A after MMR if edited strand as the repair template. Why the bystander pair A::T could be converted into G::C by error-prone polymerase?

Response: We thank the reviewer for this comment. The mechanism of deamination is reviewed by Odegard and Schatz (2006), Nature Reviews Immunology in Figure 2 (doi: 10.1038/nri1896). How the A::T converted to G::C was described by Wilson et al., (2003), Journal of experimental Medicine (doi: [10.1084/jem.20042066](https://doi.org/10.1084/jem.20042066)).

Briefly, AID initiates the deamination of C nucleotides. If the resulting U::G mismatch is not repaired before the onset of DNA replication, DNA polymerases will insert an A nucleotide opposite the U nucleotide creating C>T and G>A transition mutations. If the U nucleotide is removed by uracil-DNA glycosylase (UNG), an abasic site is created, replication of which should give rise to both transition and transversion mutations. In addition, U::G mismatch recruits the mismatch repair (MMR) machinery, which is thought to create mutations at A::T near the initiating U::G lesion, probably through an error-prone patch repair process.

We added the below reference for clarity in the results section for Figure 1, Line 180, in the revised version of the manuscript.

“Odegard, V.H. and Schatz, D.G., 2006. Targeting of somatic hypermutation. *Nature Reviews Immunology*, 6(8), pp.573-583.”

3. In the line 176-177, "Once the cytidine deaminase (AID, which converts) is generated, the uracil-guanine (U- G) mismatch can be misread". Is it "once the uracil is generated" ? Even though AID is generated, C>U is generated in a very low frequency. In addition, there seems no relationship between AID generation and U-G misread. Please modify the sentence to make it clearer.

Response: We thank the reviewer for this excellent comment. We modified the sentence below in the revised version of the manuscript.

“Once the cytidine deaminase (AID) converts C>U, the uracil–guanine (U–G) mismatch can be misread, resulting in a C>T or G>A transition.”

4. In the line 147, the author described the T7 RNAP-deaminase tool "For continuous in vivo mutagenesis of target sequences several kilobases long", what is the maximum length for targeted gene with efficient base editing? Does the mutation rate decrease with increasing distance from T7 promoter?

Response: The editing window tested in eukaryotic cells is 0.5 kb – 4 kb using the TRACE (Chen et al., 2020). In bacteria, Moore et al., 2018 suggested this system to target the DNA regions of any appropriate length. In fact, they increased the number of terminator sequences at the end of the target DNA to stop the activity of MutaT7. Similarly, Alvarez et al., 2020, used crRNA/dCas9 complex to hinder T7 RNAP-BE hybrids, protecting the downstream DNA.

Theoretically, the T7 RNAP-BE fusion can transcribe and edit any length of the target sequence downstream of the T7 promoter.

Our study targeted the GFP (729 bp) on episomes and OsALS (1935 bp) stably integrated into the genome. Sometimes, it might be necessary to test the targeted sequence's length and calculate the mutation rate with a distance from the T7 promoter; however, this is beyond the scope of this paper.

5. When the authors tested T7 RNAP-deaminase in *Nicotiana benthamiana*, the most resulted mutations are C > T or G > A. However, when they applied the tool to evolve rice ALS, the most mutations generated are not them instead of other random conversions (figure 3 and figure 4). Is it likely that the mutations in ALS are caused by transcription error of T7 RNAP rather than deaminases?

Response: We thank the reviewer for this excellent comment. If the mutations are caused by transcription error, they might be on the mRNA, not the DNA.

However, to answer this question and to understand the reader, we added the below paragraph in the Discussion section of the revised version of the manuscript.

“We observed different mutation ratios for transient expression of GFP in tobacco and stable expression of *OsALS* in rice. In the transient assays of the tobacco leaves, the DNA is available on the episome, while in the stable rice transformation, the DNA is integrated into the chromosomes. The episome lacks the physiological chromatin; thus, the chromatin modifications, the binding of chromatin modifiers, transcription factors and cofactors, or chromatin accessibility are different factors which may impact the mutation rate among transient and stable assays.”

6. If T7 RNAP-deaminase strategy can result in genome-wide DNA or RNA off-target like CRISPR-based BEs?

Response: We thank the reviewer for this comment. The off-target was assessed on episome by Moore et al., 2018 and genome-wide by Chen et al., 2020. The off-targeting activity was negligible for all the studies.

We also tested the mutations on the upstream region of the GFP sequence in the tobacco transient assays via Sanger sequencing. We did not observe any mutations.

The method we used for the directed evolution of ALS in the stable rice lines is independent of the off-targeting effect. We subtracted the mock-treated sample (without AID) reads from the experimental sample (with AID) reads using CRISPR-sub (Hwang et al., 2020). The enriched mutations were later introduced into the ALS sequence and transformed into the wild-type rice to validate the herbicide resistance.

7. Since the T7 RNAP-deaminase strategy has been already reported in other organisms, what the differences from them when applied it in plant (e.g. efficiency, editing window)?

Response: It might be misleading to compare the mutation efficiencies of plants with bacterial or mammalian cell cultures. In the previous studies, the mutation efficiencies of cell cultures are calculated as i) mutations per base per generation or ii) mutations per day per kb. For the eMutaT7 system (Park and Kim, 2021), the mutation rate was ~3.7 mutations per day per kb in ~1.1 kb of the target region. For TRACE (Chen et al., 2020), the mutation rate is 1.47 C>T mutations per kb and 3.1 G>A mutations per kb for the ~2 kb target region. However, the mutation rate increased with increasing the incubation. In our tobacco transient analysis, we observed only C>T substitutions at the rate of 1.05 mutations per kb for the 0.7 kb target sequence. In the rice callus, the mutation rate was ~7-times for the single vector system and ~14-times lower for the two vector system compared to transient assays.

8. What are the drawbacks of this strategy at current stage compared with other synthetic directed evolution methods? How to improve in the future?

Response: We thank the reviewer for this comment. We have discussed the drawbacks and future improvements in paragraph lines 407-420. The lower mutation frequency is the major drawback in plant cells. This can be further improved by using dual-T7 promoters, dual base editors (ABE and CBE fused with T7 RNAP), and the use of repair factors or DNA binding domains.

September 15, 2022

RE: Life Science Alliance Manuscript #LSA-2022-01538RR

Prof. Magdy M. Mahfouz
King Abdullah University of Science and Technology
4700 KAUST
Thuwal 23955
Saudi Arabia

Dear Dr. Mahfouz,

Thank you for submitting your revised manuscript entitled "Synthetic evolution of herbicide resistance using a T7 RNAP-based random DNA base editor". We would be happy to publish your paper in Life Science Alliance pending final revisions necessary to meet our formatting guidelines.

-please add ORCID ID for corresponding author--you should have received instructions on how to do so

A. FINAL FILES:

B. MANUSCRIPT ORGANIZATION AND FORMATTING:

****It is Life Science Alliance policy that if requested, original data images must be made available to the editors. Failure to provide**

original images upon request will result in unavoidable delays in publication. Please ensure that you have access to all original data images prior to final submission.**

The license to publish form must be signed before your manuscript can be sent to production. A link to the electronic license to publish form will be sent to the corresponding author only. Please take a moment to check your funder requirements.

Sincerely,

Reviewer #3 (Comments to the Authors (Required)):

The authors have well-responded to almost all my comments and the current version was substantially improved and suitable for publication in Life Science Alliance

September 19, 2022

RE: Life Science Alliance Manuscript #LSA-2022-01538RRR

Prof. Magdy M. Mahfouz
King Abdullah University of Science and Technology
4700 KAUST
Thuwal 23955
Saudi Arabia

Dear Dr. Mahfouz,

Thank you for submitting your Research Article entitled "Synthetic evolution of herbicide resistance using a T7 RNAP-based random DNA base editor". It is a pleasure to let you know that your manuscript is now accepted for publication in Life Science Alliance. Congratulations on this interesting work.

DISTRIBUTION OF MATERIALS:

Again, congratulations on a very nice paper. I hope you found the review process to be constructive and are pleased with how the manuscript was handled editorially. We look forward to future exciting submissions from your lab.

Sincerely,
